# The taming of the weed: Developmental plasticity facilitated plant domestication

Natalie G. Mueller[1]*, Elizabeth T. Horton[2], Megan E. Belcher[1], Logan Kistler[3]

**1** Department of Anthropology, Washington University, St. Louis, MO, United States of America,
**2** Rattlesnake Master LLC, Richmond, VA, United States of America, **3** Department of Anthropology, Smithsonian National Museum of Natural History, Washington, DC, United States of America

* ngmueller@wustl.edu

## Abstract

Our experiments with crop progenitors have demonstrated that these species exhibit dramatic plasticity in key traits that are affected by domestication, including seed and fruit morphology. These traits can be altered by cultivating crop progenitors for a single season, in the absence of any selection for domesticated phenotypes. We hypothesize that cultivation caused environmental shifts that led to immediate phenotypic changes in crop progenitors via developmental plasticity, similar to tameness in animals. Here we focus on the loss or reduction of germination inhibitors in an annual seed crop because seeds with high dormancy are undesirable in crops, and also present a serious barrier to selective pressures that arise from seed-saving and planting by humans. Data from four seasons of observation of the crop progenitor *Polygonum erectum* L. suggest that the low plant density conditions of an agroecosystem trigger a phenotypic response that reduces germination inhibitors, eliminating a key barrier to further selection. The timing of the harvest can also be used to manipulate the germinability of seed stock. These observations suggest that genetic assimilation may have played a role in the domestication of this plant. More experimental work with crop progenitors is needed to understand whether or not this phenomenon played a part in the domestication of other plants, and to accurately interpret the significance of ancient plant phenotypes in the archaeological record.

> *"The unit of survival is a flexible organism in its environment."*–Gregory Bateson [1], 457

## Introduction

Crop progenitors exhibit dramatic plasticity in key traits that are affected by domestication, such as plant architecture, yield, the timing of emergence, flowering, fruit set, and seed and fruit morphology [2–8]. If these traits can easily be altered by cultivating crop progenitors for a single season, in the absence of any selection for domesticated phenotypes, then we need to rethink our models of how domestication was accomplished and consider the possibility that plasticity was an important pre-existing trait that predisposed some plants to be domesticated. The importance of behavioral plasticity for animal domestication has long been recognized. In order to be domesticated, a wild species has to include tamable individuals: those who are capable of breeding and raising young in close proximity to people without exhibiting

**Data Availability Statement:** All relevant data are within the paper and its Supporting Information files.

**Funding:** This research was funded by 1. National Science Foundation Dissertation Improvement

Grant # BCS-1360868: A case study in agricultural practice and domestication: Knotweed in eastern North America (NGM) 2. National Science Foundation Social and Behavioral Sciences Postdoctoral Research Allowance # 1714462: Growing lost crops: Variation under cultivation in the Eastern Agricultural Complex (NGM) https://www.nsf.gov/dir/index.jsp?org=SBE 3. National Museum of Natural History Research Grant Program for Science: The palette for selection: Growing and sequencing lost crops to understand the role of plasticity in plant domestication (NGM and LK) https://nhm.org/research-collections/grants-and-opportunities The funders had no role in study design, data collection and analysis, decision to publish, or preparation of the manuscript.

**Competing interests:** The authors have declared that no competing interests exist.

aggression or debilitating fear [9]. For example, evidence for provisioning and herding [10], and manipulation of herd demographics [11, 12] precedes evidence for changes in skeletal morphology of goats (*Capra aegagrus*) in the archaeological record, and these behavioral indicators are used to recognize a critical early phase of the domestication process. Failed experiments in animal domestication, such as the managed fallow deer (*Dama mesopotamica*) of Cyprus [13], may be explicable on the basis that these animals lack the necessary temperament within the range of their behavioral plasticity to thrive in close association with humans.

We hypothesize that cultivation caused environmental shifts that led to immediate changes in crop progenitors via developmental plasticity, similar to tameness in animals. Though we have no equivalent term for tameness in plants, crop progenitors have similar pre-existing developmental plasticity that may have eased their integration into anthropogenic ecosystems. Clearing, fertilizing, weeding, thinning, pruning, and many other horticultural practices altered the micro-environments where cultivated, managed, or harvested crop progenitors grew. If plants responded rapidly in ways that were beneficial to early cultivators, for example by producing higher yields, larger seeds, seeds with less robust germination inhibitors, or a second crop in a single growing season, this would have encouraged humans to continue investing in the co-evolutionary relationship, and also expanded the phenotypic targets of selection. In some cases, such plastic responses might actually have enabled new kinds of interaction with evolutionary consequences. For example, the presence of strong germination inhibitors in many annual plants presents a puzzle for theorists of domestication [14]: many of the selective pressures hypothesized to cause the evolution of the domestication syndrome of annual plants arise from seed-saving and planting [15] (Table 1). For thousands of years prior to the advent of seed industries in the 20th century, all annual seed crop farmers selected and stored seed stock, which is the portion of the harvest that is intended for planting, rather than consumption. But why would people ever start planting if the seeds they planted mostly failed to germinate [14, 16]? If seed dormancy can be reduced via developmental plasticity in crop progenitors, then the cycle of harvesting seed stock and planting could be initiated. Here we use data from four seasons of experiments with the crop progenitor erect knotweed (*Polygonum erectum* L.) to explore this hypothesis.

## Background

### Bias against recognizing the role of developmental plasticity in domestication

The phenotype includes all developmental and physiological traits of an individual. The phenotype is ever-changing throughout the life of an organism and has the capacity to alter its

**Table 1. Hypothetical selective pressures on annual plants associated with seed-saving and planting.**

| Horticultural practice | Selection for... | Relaxation of selective pressures | Domestication syndrome characteristic |
|---|---|---|---|
| Storing seed stock (seed intended for planting, rather than consumption) | ...seeds that do not need long periods of moisture and/or specific temperature changes to germinate | Decrease in seed predation, exposure to pathogens, and risk of germinating at the wrong time of year + dispersal-specific pressures, ex. endozoochoric seeds no longer have to survive digestion [53] | Loss of dormancy and reduction in structures that maintain dormancy and/or protect the seed, usually seed or fruit coat. Loss of seed/fruit polymorphism |
| Planting in places with minimal or managed disturbance (e.g. flooding, fire) | ...plants than can thrive under a constant set of conditions | Decrease in loss of entire cohorts of plants before reproduction due to natural disaster [46] | Loss of seed/fruit polymorphism and loss of plasticity |
| Tilling/Clearance Thinning and weeding | ... seeds that germinate rapidly and seedlings that grow quickly (Harlan, et al. 1973) ...plants than can thrive under a constant set of conditions | Decrease in competition with other species and with slower germinating/growing siblings | Increase in seed size, reduction in structures that maintain dormancy Loss of seed/fruit polymorphism and loss of plasticity |

own environment. Plasticity is responsiveness or flexibility in the phenotype: the ability of an organism with a given genotype to produce alternative or continuously different forms. Plasticity can be expressed through morphology, physiology, or behavior, and is not inherently either adaptive or maladaptive. In her seminal book on developmental plasticity and evolution, West-Eberhard [17] called for a focus on "phenotypic evolution." She argued that if we are primarily interested in evolution as a result of selection, as opposed to mutation or drift, then focusing on the phenotype is sensible because this is the unit that mediates interaction been genotypes and the environment.

West-Eberhard argued that most evolutionary biologists since the mid-20th century had been focused on ultimate causes of evolutionary change, by which they meant changes in gene frequencies in populations. Phenotypic variation, including developmental and behavioral plasticity, had been characterized as proximal causes [18]. Focusing on ultimate causes to the exclusion of proximal ones can be misleading because natural selection can only act on genes if they are 1) expressed, and; 2) result in phenotypic differences that affect survival and reproduction. She wrote: "Among the consequences of the neglect of mechanisms in modern evolutionary biology are the problems that arise when the black box of the mechanism is filled in with the imaginary (2003:11)." In the 20 years since, there has been explosive growth in the number of investigations into this "black box," and functional genomics and other "omics" (transcriptomics, proteomics, metabolomics) have begun to reveal the mechanics of the complex relationships between genomes and phenotypes in many crop species [19–24].

Despite this progress, models explaining the selective pressures that led to the domestication of annual seed crops have gone largely untested. Harlan and colleagues [15] argued that simple cultivation and harvesting techniques, such as thinning seedlings, increasing planting depth, or harvesting using a sickle, acted as selective pressures for common aspects of the domestication syndrome of annual plants. Hillman and Davies classic study [25] is a rare example of an attempt to demonstrate the plausibility of this theory experimentally. The goal of the study was to determine how harvesting methods could have led to fixation of non-shattering phenotypes in wheat and barley. This research is widely misunderstood as an experimental study of selective pressures under cultivation, when in fact Hillman and Davies attempts to cultivate wild einkorn (*Triticum monococcum* L.) were failures (in their eyes). Their estimates of evolutionary change under cultivation were based on $1m^2$ harvests of wild einkorn and of domesticated emmer wheat using four different methods, which were used to build a mathematical model of selective pressures under different harvesting regimes. Hillman and Davies recognized the inadequacy of this data: "It was never intended that these preliminary data should be used statistically. . .The trials were therefore repeated on a much larger scale, using sown populations under controlled conditions with properly randomized treatments. . ."(1990:184). However, in this first experiment their shattering phenotype wild einkorn, *failed to shatter at maturity*. They moved to a greenhouse for a second try, where the wild populations never flowered. After two years of failed experiments, they decided to use their limited preliminary data rather than continuing experimental cultivation (1990: 184–5).

Hillman and Davies specifically chose to focus on a presence/absence trait (the brittle rachis) whose existence in wild populations they attributed to rare mutations, rather than, for example, seed size, which is widely understood to be affected by the growth environment. It is thus ironic and pertinent to our hypothesis that their experimental efforts were nonetheless stymied by the plasticity of a crop progenitor. They described the failure of their experiment thus: ". . .in a wet summer, brittle-rachised ears fail to disarticulate when ripe. . ." (1990:185), and went on to cite several other authors who had also noted this environmental effect in wild and semi-domesticated wheat. However, they never considered what impact plasticity in rachis disarticulation might have had during the process of domestication. Changes that occur during

domestication are not just evolutionary, they are also developmental. Because we lack the practical experience with crop progenitors that ancient people had, the effects of the environment on progenitor development have gone mostly unnoticed and unstudied. Even when these effects catastrophically impacted a laborious multi-year experiment, these authors did not recognize that developmental plasticity had *transformed a wild phenotype into a domesticated one over the course of a single growing season*. We believe that this blind spot stems from a bias in domestication studies towards viewing plasticity as noise, which is getting in the way of explaining evolutionary change through fixation of genetic variants [26].

## Weediness, adaptive plasticity, and ecological development

For most people, a weed is simply an unwanted plant. Harlan and de Wet [27] acknowledged this common definition, but called for an ecological definition of weediness that stressed a weed's ability to thrive in unstable environments. Even in the 1960s, it was nothing new to suggest that the progenitors of many of our crops are weedy in this ecological sense– Edgar Anderson [28] and Carl Sauer [29] made this case a decade earlier. But what makes an organism successful in an unstable environment? Weeds have enormous developmental plasticity, which allows them to thrive in ephemeral or rapidly changing habitats. Like our crops, our weeds have colonized ecosystems in every climate and topographic zone on Earth alongside humans–their ability to invade new territories across environmental gradients is, by some definitions, what makes them weedy.

Adaptive plasticity shapes the relative ecological breadth of closely related species, with species that are cosmopolitan weeds exhibiting a greater capacity to respond to the environment than their less widespread relatives [26]. For example, the common weed *Persicaria maculosa* Gray is able to allocate more energy to leaf biomass in shady conditions than its less common cousin *P. hydropiper* L. Opiz, which grows only in sunny locations. *P. maculosa* is also more able to allocate energy to root biomass in drought conditions than its cousin *Polygonum cespitosum* Blume, which grows only in moist soils. These studies and many others within the *Persicaria/Polygonum* study system [30–34] fall under the umbrella of ecological development, which is "the study of development as it occurs in nature, and its ecological consequences" [26]. Our research draws from this perspective and its methodologies: we are interested in development as it occurs *in agroecosystems*, and its evolutionary consequences. We also benefit from the coincidental fact that the model genus for this research (*Persicaria*) is closely related to a crop progenitor, erect knotweed, that we have been experimenting with for several years.

## Genetic assimilation and domestication

This approach has already yielded compelling evidence that developmental plasticity played a role in the domestication of maize. Piperno [35] and colleagues [5, 36] suggest that maize is a good model plant for the study of developmental plasticity and plant domestication because of genomic and transcriptomic data indicating that 1) two key domestication genes, *tb1* and *gt1*, respond to environmental cues [37, 38], and; 2) the domestication of maize involved extensive modification of regulatory elements and gene expression [39]. They explored the responses of teosinte to growth in simulated Late Pleistocene and Early Holocene conditions of reduced temperature and $CO^2$, and documented the appearance of maize-like phenotypes including apical dominance, changes in inflorescence sexuality and location, and synchronous seed maturation. They subjected the offspring of the maize-like phenotype plants to further artificial selection experiments [5] and conducted a gene expression study [36]. Differential gene expression occurred between the two treatments as expected. However, they also showed that

teosinte had greater plasticity in terms of gene expression than maize. These findings support the hypothesis that genetic assimilation played some role in maize domestication.

Genetic assimilation is the mechanism by which a phenotype that is induced as a plastic response to an environmental trigger may become fixed in a population [40, 41]. A key term for understanding this hypothetical process is the reaction norm, which is the range of phenotypes that an organism is capable of expressing in response to an environmental variable. Genetic assimilation is the narrowing of the reaction norm, sometimes called canalization [41], so that the same phenotype is expressed regardless of variation in the environment. In order for genetic assimilation to occur, the first necessary condition is the presence of a plastic trait that affects evolutionary fitness under novel conditions. This circumstance may arise either because some aspect of the environment changes, or when species disperse into new environments. While one phenotypic expression may be better adapted to the new environment than the other, genetic assimilation, which entails a loss of flexibility, would not be favored by natural selection unless there was some evolutionary cost to maintaining plasticity [40]. The same is true of selective pressures exerted by people (artificial selection). Human foragers or cultivators could have conferred selective advantages to less plastic individuals purposefully, for example by removing or not propagating individuals that expressed an undesirable phenotype. Loss of plasticity could also have evolved in response to human ecological engineering, or niche construction [42]. Since plasticity is adaptive in variable or unpredictable environments, human stabilization of the environment could relax selective pressures maintaining plasticity (Table 1).

## Erect knotweed case study

Erect knotweed was domesticated by Indigenous farmers in what is now Kentucky by c. 1 AD [43] and again by farmers in the American Bottom floodplain of western Illinois by c. 1000 AD [44]. The domesticated sub-species (*P. erectum* ssp. *watsoniae* N.G. Muell.) is now extinct. It can be differentiated from its wild progenitor in the archaeological record by 1) larger fruit (achene) size, and; 2) a loss of fruit dimorphism in favor of fruits with thin pericarps which germinate more readily [2, 45, 46]. We will use the adjective "free-living" in lieu of "wild" to refer to living erect knotweed plants on the landscape today. Free-living denotes not managed or cultivated by people. Because this species was cultivated and domesticated in the past, we use this term rather than wild to acknowledge that any of these populations could be descended from cultivated or domesticated erect knotweed. Also, erect knotweed only occurs in places that are very obviously anthropogenic, and thus are not wild by most definitions of the word. Free-living erect knotweed begins fruiting in mid-summer, at first producing only achenes with a thick, tubercled pericarp. It begins to produce fruits with thin, smooth pericarps in mid-September, and harvests collected in late autumn always include both thin and thick pericarp morphs. In contrast, the extinct domesticate *Polygonum erectum* ssp. *watsonaie* produced harvests composed of only thin pericarp fruits (Fig 1) [45].

A comparative study of erect knotweed and closely related species demonstrated that monomorphic harvests like those seen in the archaeological record do no occur in extant populations [45]. In free-living erect knotweed, Mueller observed 25–72% thin pericarp morphs in large samples (~2–3,000 analyzed fruits) from populations of free-living erect knotweed collected from mid-October to mid-November of 2014–2017 (Table 2). The highest percentage of smooth morphs observed from a free-living population was 72%, but this harvest came from plants that were scoured by flood waters in early October, and probably shed many of their early-maturing thick pericarp fruits [2]. While more work should be done on variability in free-living populations, a late fall proportion of less than 50% thin pericarp morphs is normal for free-living erect knotweed documented thus far (Table 2).

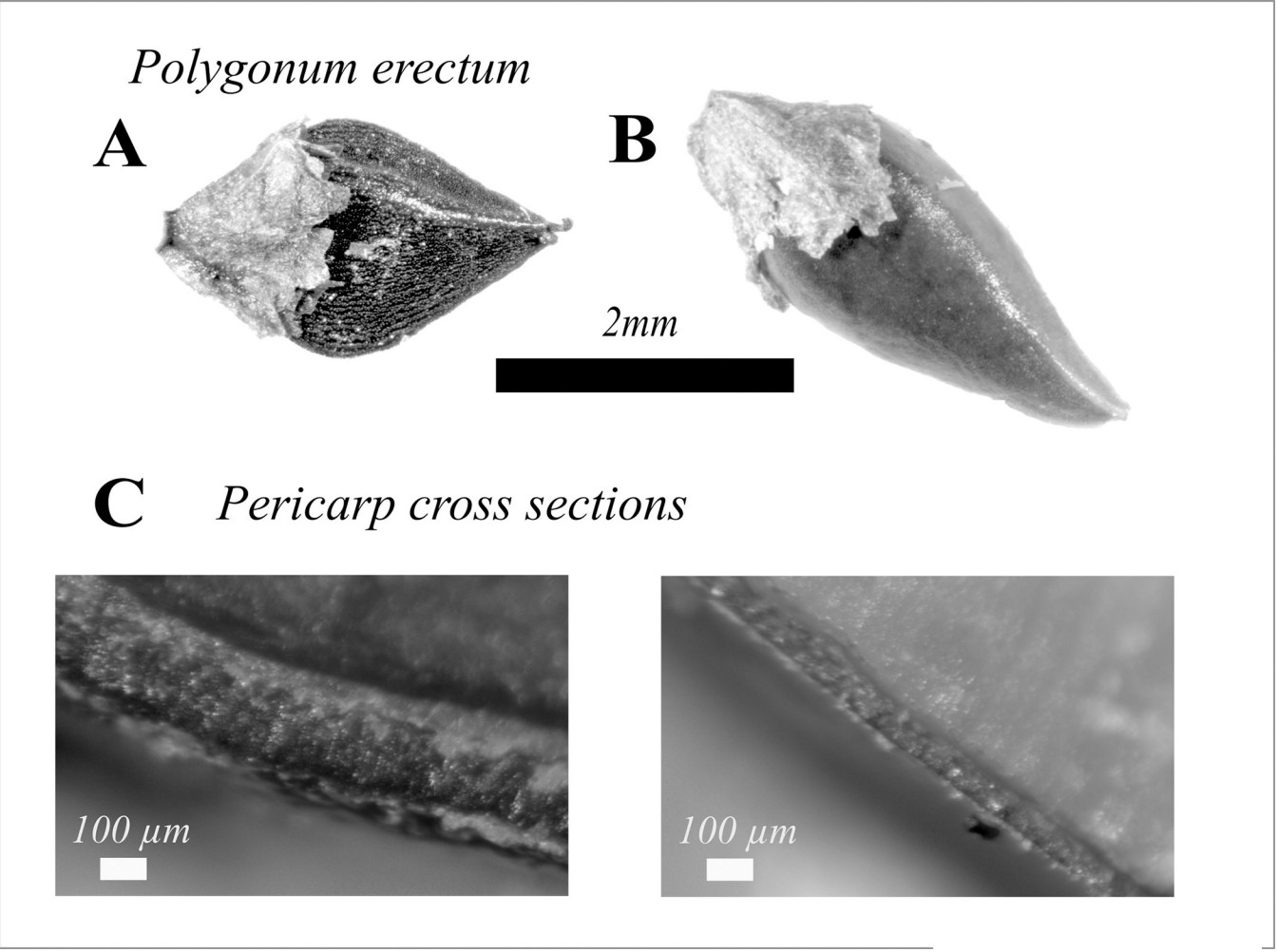

**Fig 1.** Fruit (achene) dimorphism in erect knotweed: **A**) thick, tubercled pericarp morph fruit (achene) that is >80% dormant after 6 weeks of cold, moist stratification; **B**) thin, smooth pericarp morph fruit (achene), which is >50% non-dormant after 6 weeks of cold, moist stratification, and; **C**) pericarp cross-sections, showing relative thickness of the pericarp, which restricts germination by preventing moisture from reaching the seed.

The threshold for determining whether or not an ancient assemblage shows signs of domestication based on this criterion depends on sample size, which must be sufficient to estimate whether the proportion of smooth morph seeds falls outside the range of free-living erect knotweed variation. For these experiments, we have used a sample size of 300 to estimate the thin pericarp frequency of a plant or population. Assuming a maximum wild-type frequency of 72% smooth morphs, a sample of 300 archaeological seeds with >78% smooth morphs would be outside the expected range at the 0.99 quantile under the binomial distribution, and could be interpreted as evidence of selection towards the domesticated phenotype [47]. For the experimental plants or populations reported below, we consider a percent thin pericarp morph of >78% to be evidence of a developmental difference, compared to free-living erect knotweed.

Mueller [46] demonstrated that there are significant differences between these two fruit morphs, which can be categorized using both shape and pericarp texture under low magnification (Fig 1), in terms of germination and early growth. After a six-week stratification at 4°C in moist soil, mean germination rate of thin pericarp morphs was 58%, whereas mean

**Table 2. Harvest date, plant density and frequency of thin pericarp morphs in erect knotweed harvests Free-living parent populations are shaded.**

| Year | Location | Parent population | Treatments | Unit of analysis | Replications per treatment | N | Harvest date | Plant density/m$^2$ | Mean % thin pericarp morph[1] |
|------|----------|-------------------|------------|------------------|---------------------------|---|--------------|---------------------|-------------------------------|
| 2014 | Big River, MO | — | — | Population | — | 1 | 14 Oct | High | 26 |
| 2015 | | | | | | | 22 Oct | | 29 |
| 2014 | | | | | | | 30 Oct | | 44 |
| 2015 | | | | | | | 12 Nov | | 35 |
| 2017 | Red River, KY | — | — | Population | — | 1 | 8 Oct | High | 29 |
| 2017 | Kentucky River, KY | — | — | Population | — | 1 | 11 Oct | High | 41 |
| 2014 | Crawford Creek | — | — | Population | — | 1 | 23 Oct | High | 72 |
| 2016 | Tyson Research Center, MO | Belews Creek, MO | Sun, 50% Shade | Plant | Sun: 10 Shade:14 | 24 | 18–22 Nov | 6 [54] | 70 |
| 2018a | Cornell University, NY | Belews Creek, MO | Sun, 50% Shade | 5 plants | 6 | 12 | 30 Oct | 25 | 57 |
| | | | | 25 plants | 6 | 12 | 8 Nov | | 61 |
| 2018b | | Tyson Research Center Experiment | Plant density variable | 1m$^2$ block (variable number of plants) | 1 | 9 | 7 Nov | 41 [54] | 51 |
| 2019 | Center for American Archaeology, IL | Red River, KY | Fertilizer, control | Plant | 10 | 20 | 4 Nov | 20 | 60 |
| | | Kentucky, River, KY | | | 10 | 20 | 15 Nov | 20 | 78 |
| 2020 | Tyson Research Center, MO | Cornell University Experiment | Fertilizer, deep planting, control, thinning | 1m$^2$ block (variable number of plants) | 3 | 12 | 22–25 Oct | 220 [54] | 40 |

germination rate of thick pericarp morphs was 17%. If ancient farmers were planting, as opposed to letting existing stands reseed themselves, seed stock that contained a greater percentage of thin pericarp morphs would be more likely to germinate the year it was planted. This experiment also showed that a) growth of seedlings from thin pericarp morphs was more rapid, in terms of both plant heigh and number of leaves, over the first five weeks after emergence, and; b) yield per plant was negatively correlated with plant density. Plants growing at lower densities had more axillary branches, and, since seeds are produced in the axils of branches, highly branched plants were produced more seed. Based on these results, Mueller argued that ancient farmers observed the relationship between density and yield, and thinned stands of erect knotweed at some point during the growing season. If this occurred during early growth, their thinning would have favored lineages that produced more thin pericarp seeds, since these seedlings grow faster [46].

Domestication is usually defined in terms of fixed phenotypic or genotypic changes as a result of novel selective pressures imposed by people. Most of the observations made in this initial experiment with erect knotweed readily lend themselves to a traditional narrative of domestication: Mueller proposed a combination of the practice of thinning and the relaxation of selective pressures that maintained seed dimorphism in the wild as explanations for the evolution of domesticated erect knotweed [46]. However, upon examining the morphology of the fruits produced in this first experiment, Mueller found that they had produced *no thin pericarp morphs*, instead producing a monomorphic harvest that was the exact inverse of the domesticated type. Erect knotweed plants in this experiment were grown under an artificial light regime in a greenhouse (constant day length of 16 hours), suggesting that the development of thin pericarp morphs is triggered by changes in day length in the fall.

Previous work by Sultan [30] with a related species (*P. maculosa*) showed that plants grown in shady conditions produced fruits with thinner pericarps than plants grown in full sun, suggesting that light availability in general might influence pericarp thickness of offspring. Sultan

had used light availability (sun/partial shade) as a proxy for plant density in her experiments with *P. maculosa*. She and her colleagues have argued that the thin pericarps of the plants grown in shade were a form of adaptive transgenerational plasticity: mother plants were signaling the crowded environment to their offspring and enabling them to germinate and grow rapidly [48]. Before undertaking these experiments, Mueller hypothesized that if *P. erectum* also responded to shade by producing thin pericarp fruits and to sun by producing thick pericarp fruits, then human ecosystem engineering might actually have worked *against* the expression of a domesticated phenotype, since the creation of sunny, garden environments with reduced competition would tend to increase the number of thick pericarp fruits produced [46]. This hypothesis inspired the 2016 and 2018a experiments reported in Table 2, which were designed to assess if shade had the same effect on *P. erectum* that it does on *P. maculosa*.

Harvests with more thin pericarp morphs would have been beneficial to farmers because they contain more nutritive material, are easier to process and digest, and especially because they germinate more reliably [46]. If farmers were able to collect harvests from wild plants with a high percentage of thin pericarp morphs, they could have planted them in late winter or early spring and been rewarded with decent germination rates, initiating a relationship that could lead to selection for a fixed domesticated phenotype (Table 1). We draw on observations from four years of experiments (Table 2) to explore the variables that effect the preponderance of this phenotype in erect knotweed harvests.

## Hypotheses

Observations collected between 2016–2020 can be used to investigate whether or not the loss of fruit dimorphism in favor of thin pericarp fruits, described as part of the domestication syndrome of erect knotweed, can be induced as a plastic response to a novel growth environment: the agroecosystem. Specifically, we explore the interactions between two farmer choice variables, harvest date and plant density, on a key phenotype affected by domestication: frequency of thin pericarp morphs in the harvest.

*Hypothesis 1*: Erect knotweed only begins to produce thin pericarp morphs in early fall and experiences a bout of simultaneous flowering/fruiting just before senescence, which can occur anytime from mid-October to late November. The fruits that develop during this last flowering are mostly thin pericarp morphs. Thus, we hypothesized that harvests collected later in the season or from plants that are more senesced will contain a higher frequency of thin pericarp morphs.

*Hypothesis 2*: Sultan's (1996) experiment with a closely related species suggested that shade, as a proxy for crowding, caused *P. maculosa* plants to produce seeds with thinner pericarps. Herman and Sultan [48] went on to argue that this was a case of adaptive transgenerational plasticity: mother plants were preparing their offspring for a crowded environment by conditioning them to germinate and grow rapidly. If this same phenomenon is also true for erect knotweed, it could mitigate against domestication because thinning of populations would signal mother plants to *decrease* the prevalence of quick-germinating seeds needed for planting to take hold as a human practice. Nevertheless, we hypothesized that, like its cousin *P. maculosa*, erect knotweed grown in the shade and/or at greater plant densities would produce more thin pericarp morphs as an adaptive response to crowded growth environment.

## Materials and methods

Data are drawn from four different experiments using three different parent populations, which are described below and summarized in Table 2. These experiments were each designed differently and had different goals, so we characterize this study as observational. Each

experiment represents a different agroecosystem with known characteristics, which provides comparable observational data on at least three variables needed to test our hypotheses: harvest date, plant density, and percent of thin pericarp morph in the harvest (Table 2). For all experiments, seeds were subjected to 6 weeks of stratification at 4˚C in moist soil before planting. Plots were weeded throughout the growing season to minimize interactions with other plant species. Determining when to harvest erect knotweed can be difficult. Since their senescence seems to be controlled more by moisture than day length or temperature, they can persist in flowering until the end of November in a wet fall. In all experiments, the decision about when to harvest was made on the basis of plant senescence, as judged by leaves turning golden to red and falling off (Fig 2A). However, in some experiments some plants had to be harvested before they had completely senesced because snowfall was imminent.

Unless otherwise noted, harvests were conducted by cutting off entire plants at ground level (excluding the roots), and plant density/m$^2$ was recorded at harvest (Fig 2D). In Table 2, plant density/m2 is given as either single value, for experiments where plant density was held constant in all treatments, or as a mean, for experiments where it varied. In the latter case, individual plot densities were used in the analysis. Plants were bagged by unit of analysis (either individual plant or plot, depending on the experiment, see Table 2) and dried indoors for a period of 2–3 days. After drying, seeds were hand stripped, then sieved though 2.8 mm mesh to remove larger leaves and branches. Residue was added to the dried plants, and the weight was recorded as non-seed biomass for an overall measurement of plant size. Dry seed was then threshed by rubbing it over a metal screen, then winnowed using a custom-built vacuum winnower to remove remaining chaff. The clean seed weight was recorded as yield. A random 1.5 mL sub-sample of the total yield was taken using a riffle splitter, then 300 seeds were sorted into thin and thick pericarp morph categories using a dissecting microscope at 10X magnification. This categorization was made by considering both pericarp texture (smooth vs. tubercled) and pericarp shape. Thin pericarp morphs have a higher aspect ratio (see Fig 1) [2].

## Collection of seed from parent populations

Seed was collected from parent populations in 2014, 2015, and 2017. In all cases, seed was collected by hand-stripping from at least ten different plants (or all plants if there were less than ten total) within a free-living population. Yield and biomass data were not collected because this would have necessitated harvesting all the seed and uprooting plants from rare populations. However, there is an obvious qualitative difference in plant size and architecture between free-living plants and experimental ones grown at low densities: free-living plants are both smaller and less branched (Fig 3).

Parent population seeds were dried and stored in paper coin envelopes in a freezer. Seed stock for three experiments came directly from free-living parent populations, while seed stock for the other two came from previous experiments (Table 2). Free-living parent populations are not included in analyses of the effects of plant density on achene morphology because density per meter data was not collected systematically, but all were relatively high-density ecosystems (compared to low-density experimental treatments) dominated by annual plants (Fig 2B).

## 2016 experiment

Seedlings were started in a hoop house at Tyson Research Center in Jefferson County, Missouri for four weeks. A total of 234 seedlings were transplanted into two outdoor treatments, full sun and 50% shade on June 7, 2016 (Fig 2C). Seedlings were initially planted at different intervals (25, 50, and 75 cm) to investigate the effects of plant density, but many plants were

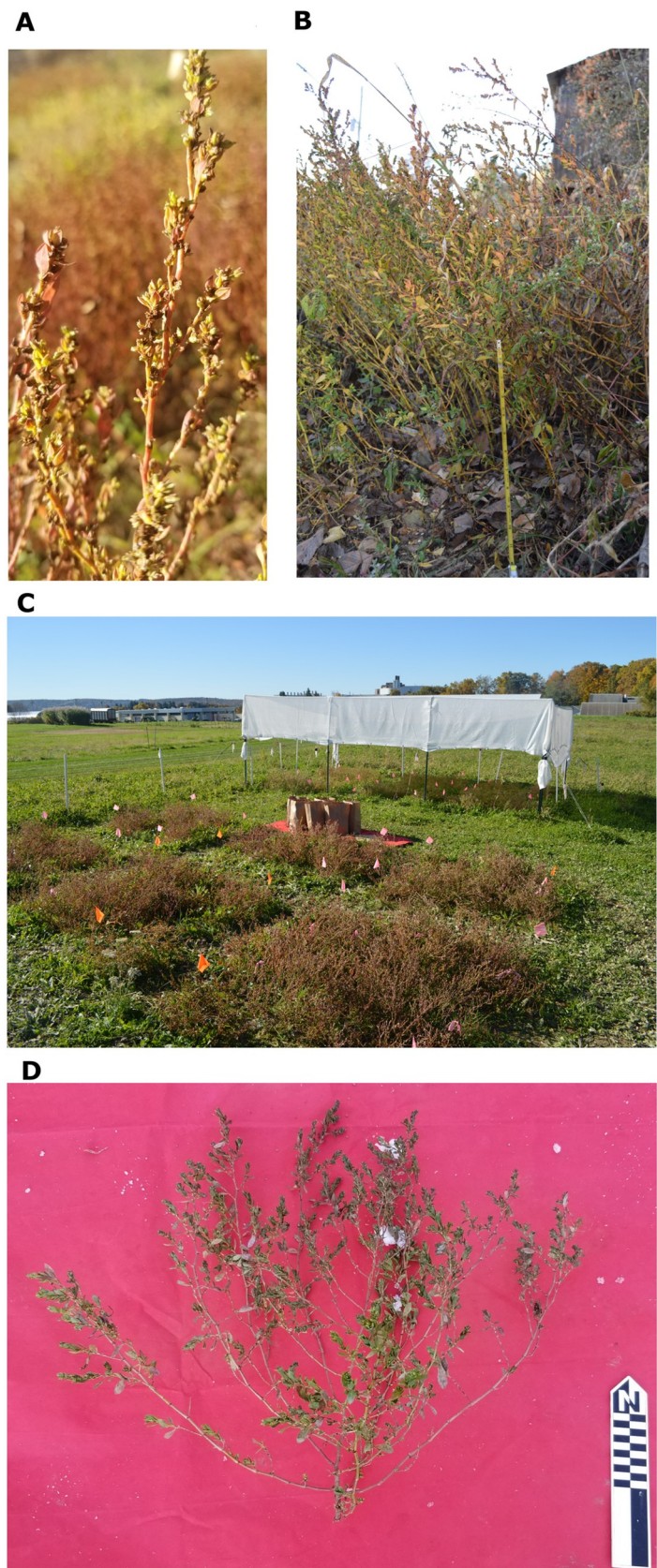

**Fig 2. A**) A typical senesced plant, example from the 2020 experiments, harvested at Tyson Research Center, MO, on October 23; **B**) a typical high-density stand of senescing, free-living erect knotweed along a dirt farm road, Big River, MO, October 22, 2014; C) 2018a experiment on October 30, shade treatment used in 2016 and 2018b experiments in background, and; D) an example of a plant that was harvested before senescence because heavy snow fall was predicted, 2019 experiment, November 15.

lost due to a drought the week after the transplant, which killed many more seedlings in the sun treatment. Ultimately, 85 plants survived until harvest in the shade treatment and only 17 survived in the sun treatment.

Plants were harvested between November 18–23, very late in the season, and the phenological stage of each plant was recorded at harvest (senesced or not senesced). We lack plant density per square meter data for this experiment, but the stem-to-stem distance to the nearest and $2^{nd}$ nearest other plant were recorded at harvest. We used this to estimate plant density per square meter for comparison to other experiments (25cm/25cm = 10 plants per m$^2$, 25cm/50cm = 7 plants per m$^2$, 50cm/50cm = 5plant per m$^2$, 50/75 = 2 plants per m$^2$). Because so many more plants survived in the shade treatment, only a stratified random sub-sample was analyzed for fruit morphology (Table 2).

## 2018 experiments

In the 2016 experiment individual plants were the unit of analysis. This presented a problem for interpreting our results because conditions for each replicant varied based on how many of its neighbors succumbed to drought. In the follow-up 2018 experiment at Cornell University in Ithaca, NY, thirty-six plants were planted in six randomized block replications of 1.25m$^2$ (25 plants per m$^2$) in each of the two treatments, sun and 50% shade (Fig 2C). Seedlings were transplanted on June 25, 2018 at a constant density 1 plant every 25 cm. An initial harvest of 5 plants per replication was carried out on October 30. For this harvest, the most senesced plants were chosen. On November 7, 25 plants in each treatment were harvested, and these were a mix of senesced plants and plants that still had some green leaves. This harvest date was chosen because heavy snowfall was imminent. Fig 2C shows the layout of this experiment and the degree of senescence of these plots before the November 7$^{th}$ harvest. We refer this this experiment as 2018a.

Simultaneously and in the same field, Mueller was also conducting an experiment to understand the effects of plant density and polyculture vs. monoculture on yield. Details on the methods and conclusions of this study were reported in 2019 [4]. Here, we use data collected from erect knotweed monoculture treatments for comparison with the 2018a experiment, because mean plant density was somewhat higher in this experiment (mean = 41 plant/m$^2$, range = 21–69 plant/m$^2$), but they were harvested at the same time and experienced identical growing conditions. We refer to this experiment as 2018b.

## 2019 experiment

In 2019, Mueller and Belcher conducted a common garden experiment using seed from two different parent populations collected in Kentucky. Ten seedlings from each parent population were transplanted into split blocks in two treatments, a raised bed containing compost and unamended soil, on June 6, 2019 at the Center for American Archeology in Kampsville, Illinois. Plants in this experiment were grown at a constant density of ~17 plants per square meter (blocks were 1.2m$^2$ and contained 20 plants). Harvests were collected and processed using the same methods as in 2018, one population on November 4 and the other on November 15 (Table 2). As in 2016, data was collected by individual plant and information on the phenological stage of each plant was recorded (senesced or not senesced).

### 2020 experiment

In 2020, Mueller conducted an experiment that was designed to investigate the effects of simple cultivation techniques on wild stands of erect knotweed in the earliest phases of domestication. Rather than transplanting seedlings in late spring, as in previous experiments, Mueller spread 10g of pre-stratified seed over 12 1m$^2$ blocks on April 17, 2020. Blocks were divided in four treatments with three replications of each: fertilizer, thinning, deep planting, and control. Cow manure was added to the fertilizer treatment blocks before planting. Seeds were covered in approximately 5 cm of soil in the deep planting treatment, whereas in the other treatments they were only covered in <1cm of soil. For the thinning treatment, Mueller thinned plots to a density of 100 plants/m$^2$ on May 29, 2020, approximately 4 weeks after emergence, favoring the largest seedlings. Plots were kept free from other species but were not thinned further, resulting in plant densities that were similar to those in free-living populations, ranging from 80–336 plants/m$^2$, but were lower in the thinning treatment blocks than in the other three treatments. Harvests were collected and processed using the same techniques as in previous experiments, between October 22–25, 2020 from mostly senesced plants (Fig 2A is a typical example), though the thinning treatment blocks were noticeably more senesced than the others. As in 2018, data was collected per block, rather than per plant so individualized data on plant phenological stage is not available.

## Results

First, we compared our data on percent thin pericarp morphs in free-living populations to our experimental datasets to see if the agroecologies that we created had any effect on this phenotype overall. We found that cultivation of erect knotweed does significantly affect this key domestication phenotype. The percent thin pericarp morph was significantly lower (two-tailed t-test, p = 0.0069) in free-living populations (n = 7, mean = 39%, Std. Dev. = 15.8%) than in experimental plants/populations (n = 122, mean = 63%. Std. Dev. = 14.2%). Some of the harvests taken from experimental plants had sample proportions of thin pericarp morphs outside the range of variation documented for free-living plants. As discussed above, a percentage of thin pericarp morphs > 78% is atypical for a free-living population (see also Mueller 2017c). The highest percentage of thin pericarp morphs we observed was 92%, from an individual plant in the 2016 experiment. Overall, there were 20 harvests that yielded sample proportions >78%, and all came from 2016 or 2019 experiments, meaning they were taken from individual plants, not entire plots (Table 1). This is significant because, based on the current published criteria, these harvests would be classified as semi-domesticated if they were recovered from an archaeological context.

### Hypothesis 1. Harvest date, senescence, and percent thin pericarp morph

There was no apparent relationship between harvest date and percent thin pericarp morphs in free-living populations (n = 7, R$^2$ = 0.03) [45, 46], but after combining data from free-living populations with those from our experiments, a relationship becomes apparent (n = 129, r$^2$ = 0.42, p<0.001, Fig 4). Among the 20 plants that produced harvests with more than 78% thin pericarp morphs, all but one was harvested in mid-late November (Fig 4). Samples with few thin pericarp morphs (<50%, n = 32 plants or plots) were more mixed: they included all but one of the parent populations regardless of when they were harvested, plants or plots from every experiment, and were harvested between October 8 and November 19. All 12 of the plots in the 2020 experiment had sample proportions of <50%; these plants were also harvested relatively early (Oct 22–25), though at this time plants were mostly senesced. Unexpectedly, for experiments where individual plant senescence was recorded at harvest, harvests from

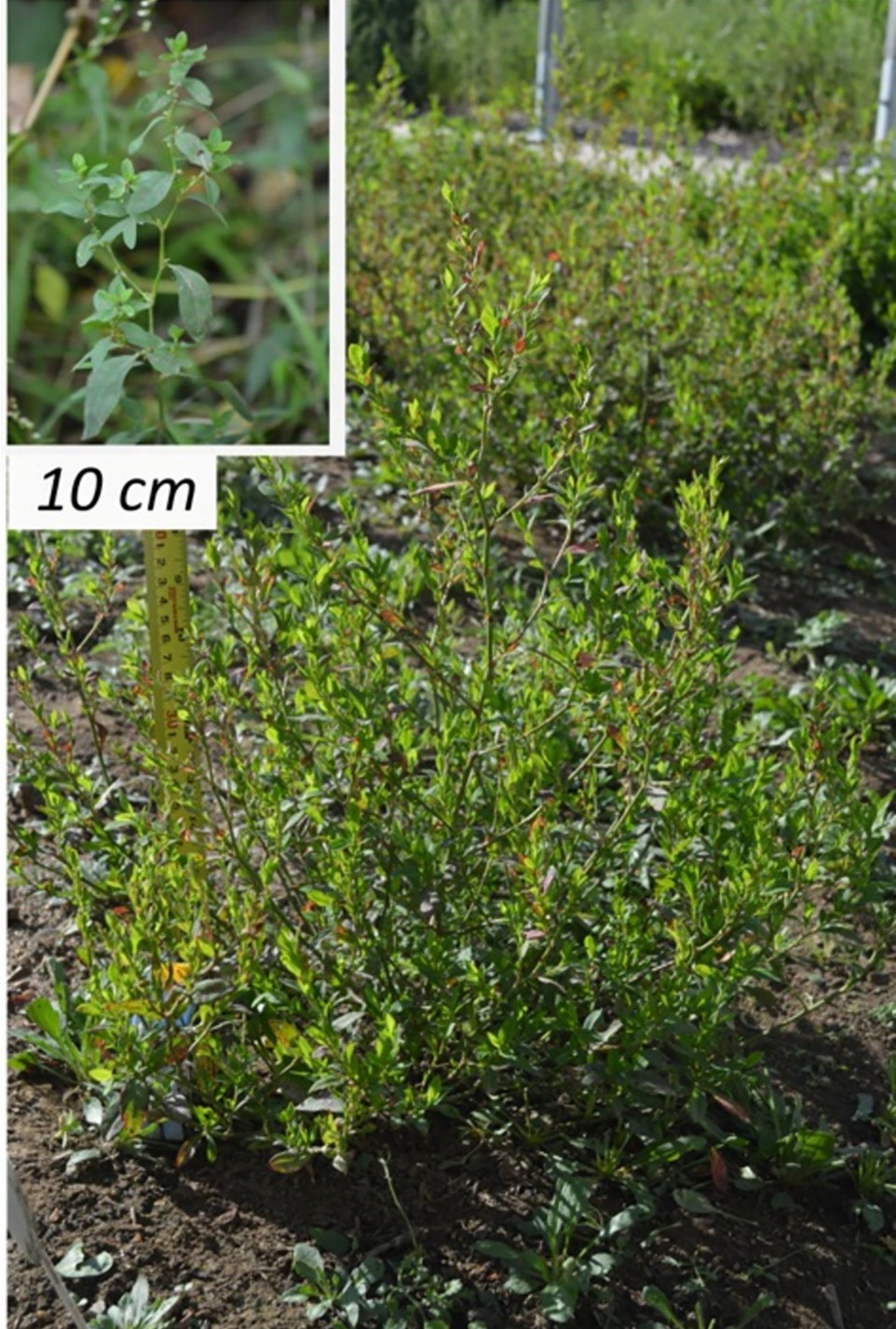

**Fig 3. Examples of differences in plant size and architecture between typical individual free-living (inset) and experimental erect knotweed plants.** Images are to scale.

completely senesced plants actually had less thin pericarp morphs than plants harvested before senescence (senesced n = 30, mean = 61%, Std. Dev. = 13.9%; not senesced n = 65, mean = 67%, Std. Dev. = 14.9%; Fig 4).

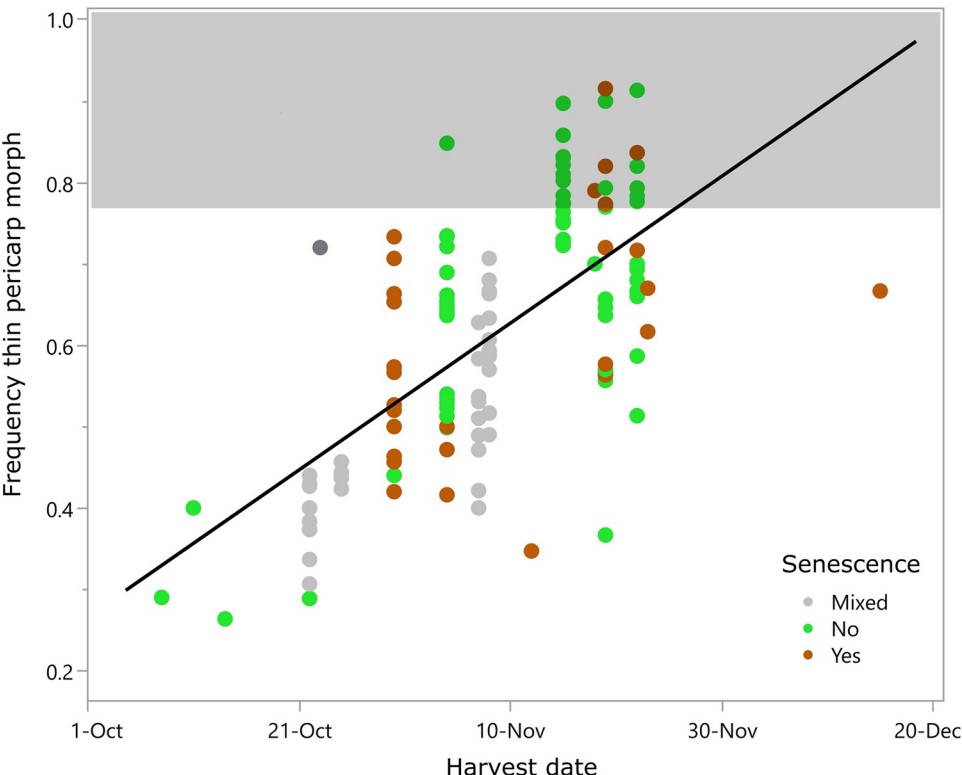

**Fig 4. Correlation between the percent thin pericarp morphs in the harvest and the harvest date with line of best fit.** Points within the shaded area are outside the range of variation documented for free-living erect knotweed.

## Hypothesis 2: Plant density and percent thin pericarp morph

Our first two experiments (2016 and 2018) were designed to follow up on Sultan's (1996) observation that pericarp thickness was greater in *P. maculosa* plants grown in sun than in shade. We did not find evidence for this phenomenon in erect knotweed: across both experiments, the mean for plants grown in the shade was 65% (n = 40, Std. Dev. 12.7%), and the mean for plants grown in the sun was 66% (n = 20, Std. Dev. 10.9%). It was this negative result that prompted us to include comparable data from other experiments. This decision converted our study from experimental to observational, but allowed us to explore other factors that might explain variability in this phenotype. Sultan and colleagues used shade as a proxy for a crowded growth environment. When we consider plant density directly (plants/m$^2$) across our experiments, we see that there is a relationship between density and pericarp thickness, but it is the opposite of what Sultan inferred for *P. maculosa*: the less dense the plot, the higher the frequency of thin pericarp morphs produced (n = 122, r$^2$ = 0.26, p = <0.0001; Fig 5). We went on to group our experimental data into naturalistic (similar to free-living populations that we have observed) and low-density groups, using 50 plants/m$^2$ as the cut-off. Harvests from low-density plots produced significantly more thin-pericarp morphs (n = 107, mean = 65%, Std. Dev. = 12.6%) than from naturalistic ones (n = 15, mean = 43%, Std. Dev. = 7.1%) (Welch's t-test, p<0.0001; Fig 5).

## Limitations

We initially hoped to used shade to simulate crowding in order to replicate Sultan's experiment with a related species. We designed two seasons of experiments (2016 and 2018a) to

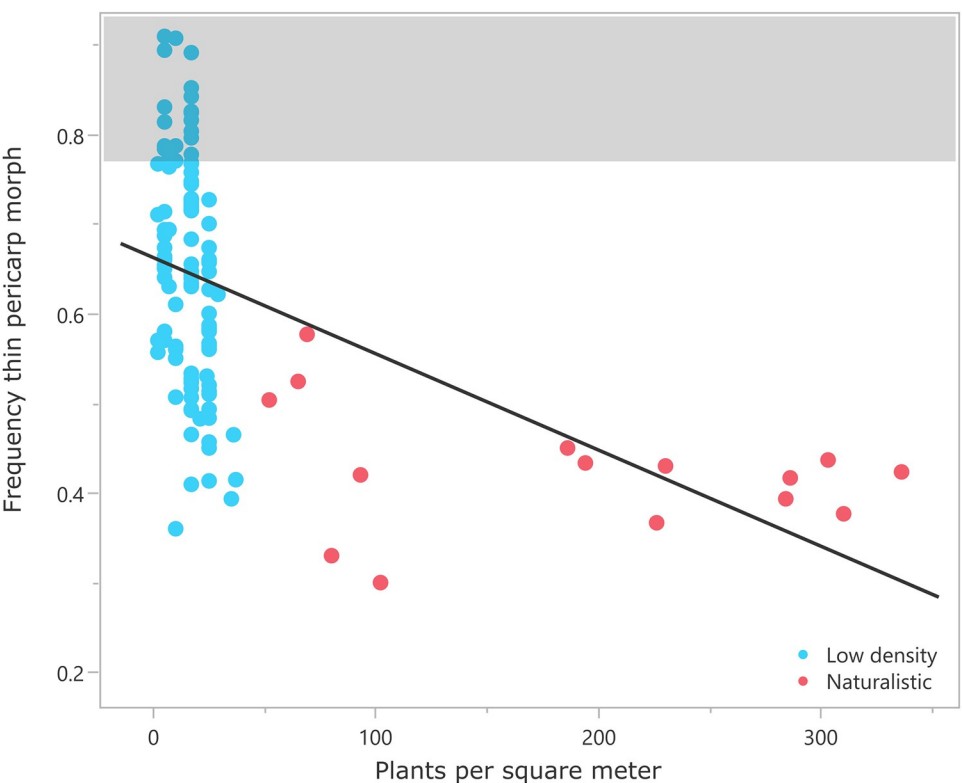

**Fig 5. Correlation between the percent thin pericarp morphs in the harvest and plant density with line of best fit.**
Points within the shaded area are outside the range of variation documented for free-living erect knotweed.

detect a difference between these two treatments, but ultimately did not find one. By combining this data with observations from other experiments, we find evidence that both plant density and harvest date have an effect on the percentage of thin pericarp morphs in the harvest. However, the experiments we synthesize here were not designed to test the effects of these variables, so our data have important limitations.

Unfortunately, all but 3 of our naturalistic, high-density plots were also harvested relatively early (Oct 22–25), and many the plants grown at the lowest densities (2016 and 2019 experiments) were also harvested relatively late (Nov 4–22) (Table 1). Since we made decisions about when to harvest based on when most plants had begun to senesce, this might indicate that plants senesce later when grown at low densities. On the other hand, in the 2020 experiment we observed that the plants in the thinning treatment were more senesced at harvest than those in other treatments, which were higher density. The relationship between density and senescence timing cannot be resolved with this dataset because harvest dates are also not directly comparable across different years and experimental sites, especially between Missouri and Illinois (2016, 2019, and 2020 experiments) and New York (2018a and b experiments). For context, the average high temperature on November 1 in St. Louis, Missouri is 17°C, whereas the average high temperature in Ithaca, New York on that day is 12°C. So, while we can say that both harvest date and plant density appear to have an effect on harvest composition, we cannot control for either using the entire dataset.

By limiting the analysis to only the 2018a results, we can control for density and all other variables and compare the harvest composition of plants harvested on Oct 30 to those harvested on Nov 8 from the same experimental treatments. There is not a significant difference

in mean percent thin pericarp morph between these time points (two-tailed t-test, p = 0.2336; Oct 30, n = 12, mean = 57%, Std. Dev. = 10.3; %Nov 8, n = 12, mean = 61%, Std. Dev. = 6.5%). In this experiment, a few remaining plants were left in the field to study seed retention over the winter. On December 15, a thaw allowed Mueller to collect one additional seed morphology sample from the remaining plants in this experiment, a snapshot of the percent thin pericarp morph more than a month after senescence. This harvest contained 67% thin pericarp morphs, not unusually high compared with the plants harvested more than a month earlier. These results, considered alongside the fact that there is no apparent relationship between harvest date and harvest composition in free-living populations, suggests that plant density is *at least* also playing a role in determining the proportion of thin pericarp morphs in the harvest.

There is one additional factor to consider. Plants in the 2016, 2018a, and 2019 experiments produced more thin pericarp fruits than did free-living populations or plants in the 2018b and 2020 experiments (Table 1). The former group used seed stock that was the first generation harvested from free-living plants (generation 1), whereas the latter used seed stock that was produced in previous experiments (generation 2). If we group the data in generation 0 (free-living populations), generation 1 (2016, 2018a, and 2019) and generation 2 (2018b, 2020), we see that the percentage of smooth morphs differs significantly between these groups: generation 1 has a significantly higher mean percent thin pericarp morph than either generation 0 or 2 (Tukey test, p>0.0001 for both comparisons). If, as Sultan (1996) contended, maternal tissues like the pericarp are a mechanism by which the mother plant signals something about her growth environment to her offspring, then the growth environment of the parent population should not be ignored.

## Discussion and conclusions

We began by hypothesizing that aspects of the domestication syndrome of annual plants can be induced in wild plants by manipulating their growth environment. We chose to focus on differences in seed dormancy caused by fruit dimorphism (thick/thin pericarp) in the crop progenitor erect knotweed. An increase in the percent of thin pericarp fruits in a harvest would have been immediately beneficial to farmers and encouraged a deepening of a domesticatory relationship because these fruits germinate more readily than their thick pericarp siblings. We found that growing erect knotweed in a garden environment affects the frequency of thin pericarp achenes in the harvest: plants from our experiments overall produced significantly more thin pericarp morphs than do free-living plants. We also found a significant correlation between the percent of thin pericarp morphs in the harvest and two variables under farmer control: harvest date and plant density.

The effect of harvest date is not very surprising, since studies of free-living erect knotweed have shown that it only begins to produce fruits with thin, smooth pericarps in the fall, and does so disproportionately during its final bout of simultaneous fruiting. Farmers could have manipulated this pre-existing variation, knowingly or not, by harvesting seed stock from plants very late in the season. It is also possible that the timing of senescence co-varies with plant density, with plants grown at low densities senescing later in the season (and thus also producing a higher proportion of thin pericarp morphs) than those grown at high densities, but given the variation in weather between years and climate between sites, this relationship needs to be investigated further.

Farmers could also influence the germinability of their seed stock by manipulating plant density in managed or cultivated populations of erect knotweed. Plants grown at low densities produce more thin pericarp morphs than do plants grown at high densities. Merely by thinning stands of wild erect knotweed, farmers could have caused a phenotypic change that is

usually considered a part of the domestication syndrome of annual plants (reduction of germination inhibitors). The fruits of such a harvest could be planted with much greater success than those harvested from a high density, unmanaged population. Previous experiments showed a readily observable relationship between yield and plant density, with plants grown at low densities producing higher yields. If ancient farmers noticed this phenomenon and followed the common horticultural practice of thinning seedlings by culling the smallest and leaving the largest, then thinning would have conferred a selective advantage on plants that produce a lot of thin pericarp seeds because these germinate and grow more rapidly [46]. This study suggests that thinning would simultaneously confer a selective advantage on plants that produce a high proportion of thin pericarp morphs *and* create a growth environment that encourages this phenotype in the next generation.

It is also possible that erect knotweed began its relationship with people as a weed in gardens or fields of other crops. It was domesticated nearly 2,000 years after goosefoot (*Chenopodium berlandieri* ssp. *jonesianum*), sunflowers (*Helianthus annuus*), and sumpweed (*Iva annua* var. *macrocarpa*) [43, 49]. If this was the case, then low density populations of erect knotweed might have at first resulted from imperfect weeding, and then progressed into a managed ground cover below these taller crops. This polyculture would be functionally similar to the three sisters polyculture used by Indigenous people in eastern North America later, in which squash was used to suppress weeds around maize plants. Previous experiments have demonstrated that a polyculture of goosefoot and erect knotweed is both easier to maintain and higher yielding than either grown as a monoculture [4].

While our experimental plants did not exhibit the full domestication syndrome that has been documented in ancient erect knotweed, which includes both larger seeds and completely monomorphic harvests, they did exhibit what might be interpreted as evolution towards this phenotype in the archaeological record, if plasticity was not considered. Yet in our experiments, no selection towards a domesticated phenotype had occurred. This suggests that experimental data on the reaction norms of wild progenitors in anthropogenic ecosystems are necessary in order to make accurate interpretations from ancient plant and animals remains. If we can induce the appearance of a 'semi-domesticated' phenotype in a single generation with no selection, then such phenotypes in the archaeological record are direct evidence of the growth environment, not of multigenerational selection towards domestication. Phenotypes that arise in response to anthropogenic environments may be confused with domesticated phenotypes, as in our case study, or they may diverge from them. For example, Harbers and colleagues [50] raised wild boars in captivity and observed the effects of this environment on their skeletal morphology compared to free-ranging wild boars. Contrary to their expectations, they found that the boars raised in captivity were actually stronger and more robust than their free-ranging counterparts. So while domesticated pigs ultimately evolved to be weaker and smaller than wild boars, the earliest *tamed* boars may actually appear especially robust in the archaeological record. Experiential learning among progenitor species may reveal many such opportunities to directly infer aspects of ancient anthropogenic ecosystems from the phenotypes of early tamed plants and animals. Wild progenitors may also be developmentally and behaviorally responsive to aspects of the environment that have nothing to do with human activity, such as $CO_2$ availability and temperature [6], and water availability [7, 25]. In order to understand ancient human interactions with plants and animals, we also have to understand plasticity in response to ancient climates and no-analogue ecosystems. The appearance of crop wild relatives today may be very different from those encountered by ancient humans, not just because these populations have been evolving for thousands of years and, in some cases, introgressing with domesticated populations, but also because of plastic responses to environmental variables.

Similar to wild animals that were capable of living and procreating in human communities, erect knotweed plants exhibit an inherent flexibility that enabled their evolutionary relationship with people to begin. In cases where the wild progenitors of domesticated animals are not extinct, observational studies have sometimes shown variation in development or behavior that selection could have worked on. For example, donkey domestication has been seen as an aberration because wild asses are solitary, territorial, and aggressive. They have no social structure into which humans can insert themselves and are likely to react to humans with fear and aggression. But a study of wild asses in captivity [51] showed that in a small, resource rich habitat (the zoo), these ordinarily solitary creatures preferred to spend their time in close proximity to each other. Marshall and Asa also observed tentative evidence for the formation of dominance hierarchies. The authors conclude that "within the range of behavioral variability of African wild asses there are individuals . . .who respond to captivity in ways that allow them to thrive, and that could lead to domestication" (Marshall and Asa 2013:498). While it is obvious that variation is necessary in order for any evolutionary process to occur, the importance of *developmental* variation in the domestication of plants has not been adequately studied. While we hesitate to characterize the developmental shift in phenotype that we have documented as behavior, it is functionally analogous to animal behaviors that allowed for greater reproductive fitness in captivity.

Unpredictable conditions favor the evolution of dormancy polymorphism in annual plants [52], and it is clear that erect knotweed plants can vary their production of dormant and non-dormant seeds in response to environmental variables [46]. Plastic dormancy polymorphisms that arise from maternal tissues, such as the pericarp, are likely to function as a means of transgenerational signaling–a way for the mother plant to share information about her growth environment that will improve her offsprings' chance of survival [48]. Besides bet-hedging in an unpredictable environment, another possible benefit of producing dormant seeds is to avoid over-population by spreading out sibling emergence across many generations. Ladizinsky [16] argued that mass-harvesting of wild legumes by people exerted selective pressure in favor of non-dormant seeds, because it relieved the seeds left behind from competition with numerous siblings. This idea leads us to one possible explanation for the connection we observed between plant density and the proportion of non-dormant seeds in the harvest: mother plants, sensing a relatively open environment, produce more non-dormant seeds to fill it up. This is the opposite of what Herman and Sultan proposed to explain their results with *P. maculosa*. They hypothesized that mother plants, sensing a crowded environment, produced seeds that would germinate quickly in an environment of high competition. The fact that two closely related species appear to respond to crowding in opposite ways may be an artifact of their unique evolutionary trajectories, in which different strategies ultimately conferred a selective advantage–and rendered one easier to tame than the other.

We conclude that erect knotweed was likely domesticated at least partly via genetic assimilation. Removal from its floodplain habitat was probably a pivotal moment in the evolution of domesticated forms in this species. The production of both dormant and non-dormant seeds had been adaptive because it allowed a mother plant to hedge its bets. Most of the thick pericarp morphs do not germinate the spring after they are produced. When an adverse event, such as a summer flood, wipes out an entire generation of plants, the thick pericarp morphs allow the population to regenerate. Within an agroecosystem where seed stock was protected by people, this kind of bet-hedging strategy would have been both unnecessary and maladaptive [46]. Now we have also shown that the production of up to 92% thin pericarp morphs is within the reaction norm of erect knotweed in a low-density agroecosystem. The process of domestication may have narrowed this reaction norm, making it impossible for domesticated populations to go back to producing dimorphic harvests. However, it is also possible that

cultivated populations retained their plasticity throughout erect knotweed's history as a crop, and that the domesticated populations in the archaeological record are only different from modern free-living erect knotweed because of the anthropogenic environment in which they grew.

It would be nearly impossible to *prove* that erect knotweed was domesticated via genetic assimilation, because the domesticated form is extinct: we cannot conduct experiments similar to those conducted with maize and teosinte and several other crops [7, 8, 36], to see if the domesticated form is less plastic than its wild progenitor. Moreover, erect knotweed lacks genetic resources that are increasingly well-developed for many important crops. Any studies of differential gene expression underlying the phenotypic shifts we observed would have to rely on analogy to what is known about gene function in model species. We did attempt to integrate a study of gene expression into the 2018a experiment. We collected tissue samples from developing flowers and fruits every two weeks between July and November. However, since fruit morphology ultimately did not differ significantly between the treatments in this experiment, this data has limited applicability to our argument. A follow-up experiment that is explicitly designed to measure the effects of plant density on proportion of thin pericarp morph, and ideally also incorporates a study of differential gene expression between treatments, could strengthen our conclusions. With a lengthy enough longitudinal experiment, it might even be possible to experimentally re-domesticate erect knotweed. While this would not prove that genetic assimilation was involved in erect knotweed's ancient domestication, it could provide an experimental demonstration of genetic assimilation occurring in domestication in real time.

We close by calling for more experimental and observational studies of wild progenitors that attend to differences in development and behavior in varying environments. Such studies have the potential to help us understand how domestication occurred and why certain species were domesticated while others were not.

## Supporting information

**S1 File. Data on Polygonum seed dimorphism, plant density, and harvest date from all populations analyzed for this paper.**
(XLSX)

## Acknowledgments

Funding for these studies was provided by: NSF Dissertation Improvement Grant # BCS-1360868, NSF Social and Behavioral Sciences Postdoctoral Research Fellowship # 1714462, and the National Museum of Natural History Research Grant Program for Science. Experiments that involve interactions with other living beings are labor-intensive and unpredictable, and these ones would not have been possible without the hard work of many people. Paul Krautmann kindly allowed NGM to access a large population of erect knotweed on his property, which provided much of the seed used in these experiments. Don Conner helped harvest the 2016 plants. Jane Mt. Pleasant and the staff of the School of Integrative Plant Sciences at Cornell University facilitated the 2018 experiments. Tim Mueller, Andrea White, and Peter Szilagyi worked on them tirelessly throughout the summer and fall. The 2019 experiments were facilitated by the staff and students at the Center for American Archeology, especially Jason King, Jane Buikstra, and Daniel Williams. Emma Frawley and Quinn Fox helped plant and maintain the 2020 experiment. The staff at Tyson Research Center facilitated both the 2016 and 2020 experiments. Annie Frisch and Abigail Cannon assisted with threshing and

winnowing harvests. We are also grateful to seven generations of erect knotweed plants for teaching us so much.

## Author Contributions

**Conceptualization:** Natalie G. Mueller, Elizabeth T. Horton, Logan Kistler.

**Data curation:** Natalie G. Mueller.

**Formal analysis:** Natalie G. Mueller.

**Funding acquisition:** Natalie G. Mueller, Logan Kistler.

**Investigation:** Natalie G. Mueller, Megan E. Belcher.

**Methodology:** Natalie G. Mueller.

**Project administration:** Natalie G. Mueller.

**Resources:** Natalie G. Mueller.

**Supervision:** Natalie G. Mueller.

**Visualization:** Natalie G. Mueller.

**Writing – original draft:** Natalie G. Mueller.

**Writing – review & editing:** Elizabeth T. Horton, Megan E. Belcher, Logan Kistler.

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
