## [Decision Letter · Decision Letter 0]

16 Feb 2023

PONE-D-22-30960The taming of the weed: developmental plasticity facilitated plant domesticationPLOS ONE

Dear Dr. Mueller,

Thank you for submitting your manuscript to PLOS ONE. After careful consideration, we feel that it has merit but does not fully meet PLOS ONE’s publication criteria as it currently stands. Therefore, we invite you to submit a revised version of the manuscript that addresses the points raised during the review process.

As you address the reviewers' comments, we would like for you to attend in particular to:

(1) Reviewer #1's request for elaboration in the discussion of genetic assimilation regarding the role of purposeful artificial selection; (2) Reviewer #1's suggestion that you incorporate reference to the fact that non-anthropic environmental change can also affect plasticity in wild progenitors; and Reviewer #2's suggestion that you include a discussion of relevant earlier experiments with knotweed. Please also double check table referents throughout the text to ensure accuracy.

We look forward to receiving your revised manuscript.

Kind regards,

Raven Garvey, Ph.D.

Academic Editor

PLOS ONE

Journal Requirements:

2. In your Methods section, please provide additional information regarding the permits you obtained for the work. Please ensure you have included the full name of the authority that approved the field site access and, if no permits were required, a brief statement explaining why

4. "In your Data Availability statement, you have not specified where the minimal data set underlying the results described in your manuscript can be found. PLOS defines a study's minimal data set as the underlying data used to reach the conclusions drawn in the manuscript and any additional data required to replicate the reported study findings in their entirety. All PLOS journals require that the minimal data set be made fully available. For more information about our data policy, please see http://journals.plos.org/plosone/s/data-availability.

Reviewers' comments:

Reviewer's Responses to Questions

**Comments to the Author**

1. Is the manuscript technically sound, and do the data support the conclusions?

Reviewer #1: Yes

Reviewer #2: Yes

2. Has the statistical analysis been performed appropriately and rigorously? 

Reviewer #1: Yes

Reviewer #2: Yes

3. Have the authors made all data underlying the findings in their manuscript fully available?

Reviewer #1: Yes

Reviewer #2: No

4. Is the manuscript presented in an intelligible fashion and written in standard English?

Reviewer #1: Yes

Reviewer #2: Yes

5. Review Comments to the Author

Reviewer #1: This is an interesting, well-done, and exciting paper that should definitely be published. The authors demonstrate developmental plasticity leading to a phenotype with domesticated features in the wild ancestor of an important native crop plant of eastern North America simply through moving the wild forms to a cultivated plot. The paper adds to the existing evidence from experimental studies of teosinte and maize that development plasticity needs to be considered in the domestication process of other plant taxa to build a complete, robust archaeological and genetic record of domestication, including accurately identifying forms found in archaeological sites as wild or domesticated. I have a few suggestions for revision.

Regarding the evidence for genetic assimilation in maize domestication, the authors say that "this plasticity was lost during domestication because a) the environment was rendered more predictable by people, and; b) the expression of a teosinte-like phenotype was maladaptive in agroecosystems." Lorant et al. and Piperno et al. considered that genetic assimilation (GA) occurred in significant part because artificial selection was placed on the maize-like plastic teosinte phenotypes, as the phenotypes offered benefits to cultivators in way of ease of harvesting and other features, and would have been selected for. At least for maize and possibly other taxa where plasticity exists or may be found, the role of purposeful artificial selection should be emphasized more in the authors' discussion of how GA occurred.

Also, there should be attention somewhere in the manuscript to the fact that environmental changes in the absence of human influences can cause plasticity in wild progenitors of crops; this effect would probably be more important in those crops domesticated shortly after the Pleistocene ended and appear to have operated in the teosinte to maize case.

Reviewer #2: Review of Mueller et al.

There has been increasing attention of late on the role of developmental plasticity in domestication. Experimental research on teosinte grown under different levels of CO2, for example, has shown that low-CO2 conditions similar to that of the Late Pleistocene/Early Holocene induced plastic responses in teosinte that result in the expression of traits that would have made morphs with these traits more attractive to foragers - thus started them on the pathway to domestication. Here, Mueller et al. have shown how human efforts at manipulating environments and harvest schedules elicited plastic responses that may have helped kick-start the domestication of erect knotweed - one of the supposed “lost crops” of eastern North America. And while the demonstration of the role of climate change in eliciting plastic responses that set plants on the pathway to domestication is significant, this demonstration of the interplay between human eco-system engineering and receptive plant species sheds light on what is likely the most common way in which developmental plasticity shaped the process of domestication. This set of carefully controlled, and clearly outlined, experiments underscores the agency of progenitors of future domesticates in the domestication process through the ability of certain species to take advantage of the new opportunities offered by anthropogenic environments through developmental plasticity. It also highlights the corollary agency of humans in this process by recognizing and encouraging plastic responses that suit their needs. Much has been written in domestication studies about the co-evolutionary relationship that is proposed to lie at the heart of the domestication process. This study takes this line of inquiry several step further by actually demonstrating this relationship in action. Clearly this is landmark work in domestication research worth of publication in Plos One.

But equally (or perhaps more) important is the contribution this set of ingenious experiments makes to evolutionary biology more broadly. It does this by going to the heart of the recent debate over the need to revise standard evolutionary theory to acknowledge the importance of processes such as developmental plasticity and genetic assimilation that are relegated to relatively unimportant secondary roles in evolution by the Modern Synthesis - the current dominant paradigm formulated in the 1940s. Advocates of the need for an Extended Evolutionary Synthesis have emphasized the importance of developmental plasticity as a mechanism capable of causing significant population in certain plant species wide change in short time frames, in particular, when exposed to novel environments. The experiments reported on here are among the most compelling demonstrations of this phenomenon I’ve seen.

But perhaps the greatest contribution of this work to evaluating core assumptions at the heart of the debate over the need for revision of standard evolutionary theory is the insights this study provides into the process of genetic assimilation by which plastic traits become encoded into the genetic architecture of the organism. This has proven one of the most contentious aspects of the EE, greeted by great skepticism by advocates of standard theory. Yet without some mechanism for the fixation of plastically arising traits, the role of plasticity in evolution is questionable. To be clear, this study doesn’t demonstrate genetic assimilation in the same way as it shows how traits that are both adaptive to the plant and attractive to humans could arise through developmental plasticity in a very short time frame. Nor does this study claim to do so. But this research does lay-out a very plausible scenario as to how this could have happened in the case of erect knotweed. It is hoped that further experimental studies, coupled with genetic analysis, will provide this demonstration. And it would be good to hear more on how this line of inquiry might be pursued in the future.

Thus, I believe that the most outstanding contribution this landmark study makes is in its demonstration of the importance of the study of domestication process as an ideal model system for exploring some of core issues at the heart of this debate over the need for revision. This broadens the appeal of this research to a much wider audience, again making it a suitable contribution to Plos One.

I have very few suggestions for revision and believe that the manuscript could easily be published as is. I might, however, suggest a bit more on the earlier experimental work with this species that demonstrated the link between thinning plots of free-growing knotweed and changes in branch architecture and seed productivity. At the time this earlier work was done the impact of these activities on tubercle morph expression was unclear. Indeed, if I remember correctly, it seemed that the activities that encouraged the expression of these two traits that would have been attractive to human harvesters also encouraged the expression of tubercled morphs that would have inhibited the success of sown seeds in human tended garden plots. This study would seem to show that this is not the case and that these activities, coupled with later harvest schedules, would have favored seeds that would do well in such plots.

I also caught a small problem with table referents. At several points in the manuscript the authors refer to Table 1 when I think they meant to refer to Table 2.

In addition, the density column in Table 2 could use with a bit more explanation. I’m not sure what the [53] figures in a number of cells mean.

Figures showing the plants could also use a bit more explication in the heading to help the reader not familiar with growing knotweed (which I think would apply to a lot of folks) decipher what is being shown.

Not sure if underlying data are presented here as I don’t see that there is supplemental information included with the paper.

6. PLOS authors have the option to publish the peer review history of their article (what does this mean?). If published, this will include your full peer review and any attached files.

Reviewer #1: No

Reviewer #2: **Yes: **Melinda Zeder

---

## [Author Response · Author response to Decision Letter 0]

8 Mar 2023

See attached document "Response to Reviewers."

---

## [Editor Report · Decision Letter 1]

27 Mar 2023

The taming of the weed: developmental plasticity facilitated plant domestication

PONE-D-22-30960R1

Dear Dr. Mueller,

We’re pleased to inform you that your manuscript has been judged scientifically suitable for publication and will be formally accepted for publication once it meets all outstanding technical requirements.

Kind regards,

Raven Garvey, Ph.D.

Academic Editor

PLOS ONE
---

## [Editor Report · Acceptance letter]

30 Mar 2023

PONE-D-22-30960R1 

The taming of the weed: developmental plasticity facilitated plant domestication 

Dear Dr. Mueller:

I'm pleased to inform you that your manuscript has been deemed suitable for publication in PLOS ONE. Congratulations! Your manuscript is now with our production department. 

Kind regards, 

on behalf of

Dr Raven Garvey 

Academic Editor

PLOS ONE